# Digitalisation in Hospitals in COVID-19 Times—A Case Study of the Czech Republic

Jarmila Zimmermannova [1,*] , Lukas Pavlik [2] and Ekaterina Chytilova [3]

1 Department of Sustainable Development, Moravian Business College Olomouc, tr. Kosmonautu 1, 77900 Olomouc, Czech Republic
2 Department of Informatics and Mathematics, Moravian Business College Olomouc, tr. Kosmonautu 1, 77900 Olomouc, Czech Republic; lukas.pavlik@mvso.cz
3 Department of Economy and Management, Moravian Business College Olomouc, tr. Kosmonautu 1, 77900 Olomouc, Czech Republic; ekaterina.chytilova@mvso.cz
* Correspondence: jarmila.zimmermannova@mvso.cz

**Abstract:** In COVID-19 times, the healthcare system needs more financial and other resources for covering all necessary medical products and services. On the other hand, we have observed pressure on the effectiveness and optimisation of resources in hospitals and healthcare facilities. Digitalisation represents an important source of information for various levels of management in hospitals. The main aim of our research is the identification of the benefits of digitalisation of medical devices in hospitals in COVID-19 times, focusing on a case study of the Czech Republic. For our methodological approach, a literature review, data analysis, correlation analysis, and regression analysis were used. The case study presents the changes to the equipment/facilities use in years 2019 and 2020 in a selected hospital in Prague and the impact of COVID-19 on such use of resources. Management and financial issues are discussed, together with recommendations for healthcare sector management. As a result, economic benefits are represented mainly by various kinds of savings and optimisation of both processes and employees. On the other hand, it is not easy to identify all possible savings, as some of them can be in non-financial expression.

**Keywords:** digitalisation; hospital; optimisation of resources; equipment/facilities; COVID-19; benefits; savings

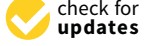



## 1. Introduction

In COVID-19 times, the healthcare system needs more financial and other resources for covering all necessary medical products and services. On the other hand, we have observed pressure on the effectiveness and optimisation of resources in hospitals and healthcare facilities. All resources should be optimized—employees, medical products, medical services, rooms, capacities. For such optimisation and decision-making, the managers need suitable data resources and well-structured data inputs. Therefore, digitalisation in hospitals can represent an important source of information for management.

Several studies have recently focused on the importance of digitalisation for modern management. Regarding general studies, Hofmann (2021) underlines that digitalisation enables new, previously unimagined, innovative business models. Companies can be faster and more flexible in developing new business ideas and new products, thanks to more direct and faster contacts and a larger amount of data. This development also leads to increasing customer demands and ever shorter reaction times.

Another study (Hoerlsberger 2019) focuses on the impact of digitisation in the industries. Based on the results, existing business models are disrupted through new technology. Thus, the author indicates that businesses today have to open up their minds to innovation and continuous learning. In the modern world, there is a definite shift towards individualised products and services as well as a unique customer experience.

Based on Gigauri (2020), management strategies need resilience, flexibility, and adaptability in digitalisation time. In the quality management (QM) area, many future digitalised solutions for improving internal work will require collaboration between functions. In this respect, the explorative-internal role of digitalisation for QM practitioners includes planning, designing, and reviewing with internal stakeholders to provide solutions that create better opportunities for the provider to offer value for the customer (Rojas et al. 2021).

Ageron et al. (2020) focused in more detail on the issues related to the digitalisation of SME supply chains and indicated that public supply chains, such as public hospitals supply chains, are under-studied.

Regarding the impacts of digitalisation on healthcare services, Iadanza and Luschi (2020) support innovative approaches in healthcare institutions. Innovation, together with digitalisation and development in health technology, contribute significantly to the quality of health care provided by various health facilities (Austin et al. 2018). Moreover, it represents new challenges for both management and staff of health care services. Health care professionals need to understand the forces that add value to the cost-effectiveness and efficiency of health care delivery systems. Based on the study performed by Blythe et al. (2019), technology and health care equipment play a significant role in health care services. It is necessary to understand the role of technology management to communicate effectively about it to health planners (Cucciniello et al. 2016). The COVID-19 situation in particular represents a big challenge for health care management and innovative approaches.

The aspects of patient safety and integration of digitalisation into the professional context necessitate an assessment of healthcare professionals' competencies in digitalisation (Basu 2020). The key competencies from a healthcare perspective include encompassing knowledge of digital technology and the digital skills required to provide good patient care, including associated social and communication skills, and ethical considerations of digitalisation in patient care (Konttila et al. 2019).

Moreover, financial benefits are linked to the use of electronic documents (Geier and Smith 2019). A good drug inventory planning system is important for efficient budgeting, procurement, and cost control of drugs. When quantities of stagnant drugs in the inventory are too much, wastage due to expired and spoiled drugs could occur. This not only causes loss of income but could also jeopardize healthcare service delivery (Dewi et al. 2020). Pouloudi and Whitley (1997) focused on drug use management information systems help to manage information on patients. Moreover, these systems can be used on drugs and their costs to monitor and evaluate the effectiveness of drug use policies (Pouloudi and Whitley 1997).

Chan et al. (2020) proposed to optimize medical expenditure by focusing on the common diagnosis segment, as this segment of individuals is one of the largest groups that drive medical claims. This analysis helps to better understand the employees' medical claims and to have an overview of the current employee health population.

Choi et al. (2013) analysed the economic effects of an electronic medical record system in hospitals that used a cost-benefit analysis based on the differential costs of managerial accounting. The benefits included cost reductions after system adoption and additional revenues both from the remodelling of paper chart storage areas and medical transcriptionists' contributions.

The digitalisation of hospitals can improve financial management (Dasgupta and Narendran 2021). Moreover, it can identify how many beds are occupied and what is the expected revenue (Kawale et al. 2020). Adopting electronic documents can save organizations money by decreasing costs associated with paper records (e.g., storage, purchasing paper, printing) and by reducing duplicate testing and other redundant interventions (Geier and Smith 2019).

Besides the economic impact of COVID-19, both public and private expenditures connected with healthcare systems are increasing for a number of reasons. Carbonaro et al. (2018) examined the links between various variables influencing economic impacts, such

as population ageing, demographic development, local economic development prospects, and financial implications. Population ageing and demographic development tend to increase the demand for health services. Burian et al. (2018) compared the consumption expenditures of households of employees with households of pensioners and identified a group of significantly higher consumption expenditures of households of pensioners. Within this group, the most significant are the health expenditures of pensioners, including expenditures on medical products, medical services, dental services, paramedical services, and hospital services.

Searching for scientific studies focused simultaneously on COVID-19, healthcare, and economic impact, we found 312 studies in total, published in the period 2020–2021 in scientific journals indexed on the Web of Science database (WoS 2021). Using the WoS tool Treemap (Wilkinson 2021) shows 10 results for scientific disciplines most connected with the keywords COVID-19, healthcare, and economy (Figure 1).

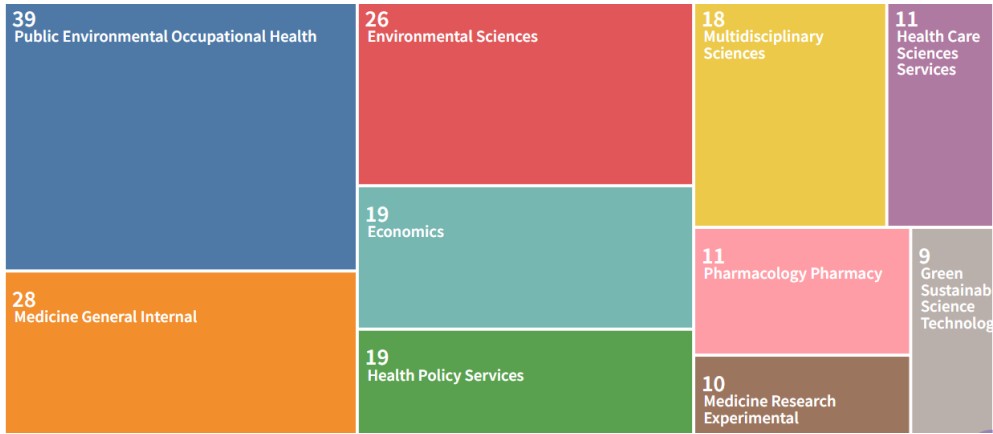

**Figure 1.** Scientific studies connected with the key words COVID-19, healthcare, and economy. Source: WoS (2021); own processing.

There are fewer studies focusing in more detail on the impact of COVID-19 on hospitals and their economic situation specifically. Using the keywords COVID-19, hospital, and economy, we found 124 relevant scientific studies (Figure 2). Regarding economic scientific studies, most of them are case studies, such as case studies from the United Kingdom (Mitha 2020), United States (Chen et al. 2020), Italy (Barbieri and Bonini 2021), France (Sainsaulieu 2021), and a complex study that presented an economic analysis for 94 countries (Vera-Valdés 2021).

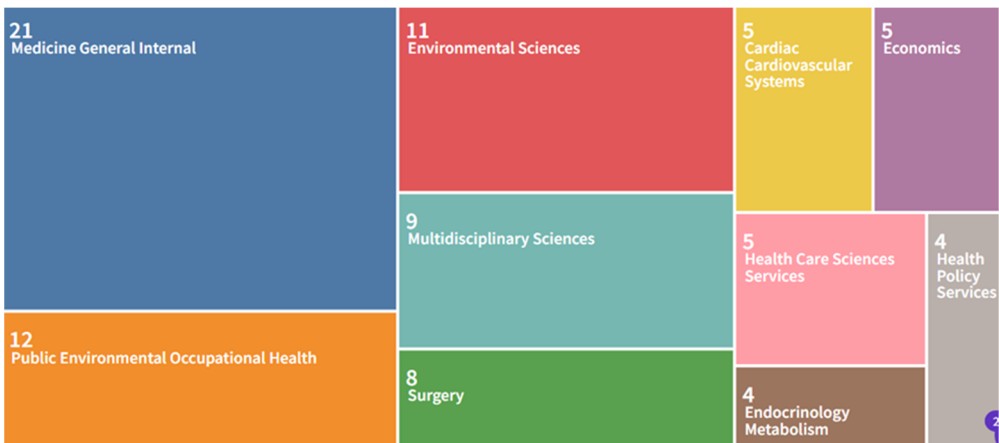

**Figure 2.** Scientific studies connected with the key words COVID-19, hospital, and economy. Source: WoS (2021); own processing.

Looking for synergies between these scientific studies, there is one common link—hospital bed/beds. Such equipment is an important indicator for management and planning in hospitals. Moreover, it is a suitable indicator for the analysis of efficiency and economic approach. For example, healthcare and hospital beds are analysed in the study performed by Barbieri and Bonini (2021), while hospital beds and staff (permanent staff, administration staff, management) are focused on by Sainsaulieu (2021) and hospital beds per 1000 inhabitants are a statistically significant factor in reducing the number of deaths (Vera-Valdés 2021). Hospital bed capacity is discussed in Chen et al. (2020). The authors considered a case where HRRs share hospital beds among the neighbouring HRRs during a surge in demand beyond the available beds and the impact it has in controlling additional deaths.

Based on these studies, "use of hospital beds" was selected as an important indicator for the correlation and regression analysis also in our study.

The paper is divided as follows: key scientific studies are described in Section 1 "Introduction", Section 2 will focus on "Methodology", the key findings will be presented in Section 3, and Section 4 of the results within the broader context of answering the research questions will follow. Finally, Section 5 underline the key findings and formulate the recommendations.

## 2. Methodology

### 2.1. Data Sources

The research presented in this paper is based on the data collected in the selected hospital, obtained from the EFAS information system (EFAServices 2021), and official data of the Czech Statistical Office (CZSO 2021) and the Ministry of Health of the Czech Republic (MoH CR 2021a, 2021b).

The EFAS information system (EFAServices 2021) is primarily intended to provide facility management for healthcare organizations in digital form. From the point of view of practical use, the users of the system are managers of buildings and medical facilities. The EFAS system also offers the possibility of CAD visualization. It is, therefore, possible to create maps of the premises and individual departments of the hospital. With the help of the EFAS information system, we can obtain a very detailed overview of individual hospital departments and the use of facilities.

The identification of evidence in medical facilities in combination with one-factor or two-factor verification of the identity of the person using the device at a particular time can help set certain regime measures. These measures can then be implemented in the safety policy of the medical organization (Pavlík et al. 2021).

We worked with data on the use of equipment/facilities, provided to us by a significant hospital located in the Czech Republic, Prague. The data was extracted from a data file related to the use of equipment/facilities. In the data file, we can identify, for example, the name of the person who used the device, the name of the department, the time of start or end of the use of the device, the type of device, etc.

Based on the filtering of selected facilities, tables were compiled in which the use of selected types of facilities in a specified time range was compared. This step in the research aimed to determine the increase or decrease in the use of facilities in a particular hospital before and during the COVID-19 pandemic. For this purpose, the time interval from the end of August 2018 to March 2021 was selected. The obtained data were then processed in graphical form, which were then used for the further research purposes of this paper.

Table 1 shows the overview of all data/variables used for research presented in this paper, including abbreviations and units of the variables.

**Table 1.** List of variables.

| Variable | Abbreviation | Unit |
|---|---|---|
| Equipment use frequency in hospital | EQUIP | Frequency of use per month (summary of equipment use below) |
| Infusion pump | INF | Frequency of use per month |
| Infusion pump ARGUS | INFA | Frequency of use per month |
| Linear dispenser | LIDI | Frequency of use per month |
| Injectomat | INJ | Frequency of use per month |
| Defibrillator LIFEPAK | DEFL | Frequency of use per month |
| Defibrillator | DEF | Frequency of use per month |
| Bed | BED | Frequency of use per month |
| Elegance bed | BEDE | Frequency of use per month |
| Lung ventilator | LUVE | Frequency of use per month |
| Number of COVID-19 positive people in Prague | COVP | Persons per month |
| Number of COVID-19 positive people | COV | Persons per month |

Source: Own processing.

Table 2 summarizes the parameters of each of the variables. The minimum and maximum values, the mean, and the median are indicated for each of the variables. For the analyses, we used monthly data (monthly averages), ensuring precisely 24 observations (24 months) for each variable.

**Table 2.** Characteristics of variables.

| Variable | Minimum | Maximum | Mean | Median |
|---|---|---|---|---|
| EQUIP | 2865 | 5791 | 4107.292 | 4076 |
| INF | 827 | 1628 | 1211.625 | 1158 |
| INFA | 110 | 272 | 198.375 | 206.5 |
| LIDI | 966 | 2448 | 1576.208 | 1578.5 |
| INJ | 280 | 724 | 500.5 | 513.5 |
| DEFL | 21 | 162 | 61.91667 | 60.5 |
| DEF | 33 | 171 | 75.33333 | 69 |
| BED | 184 | 334 | 260.1667 | 276.5 |
| BEDE | 51 | 99 | 72.875 | 71 |
| LUVE | 113 | 200 | 150.2917 | 150 |
| COVP | 0 | 13,191 | 2143.417 | 0 |
| COV | 0 | 255,830 | 26,768.13 | 0 |

Source: Own processing, based on the data EFAServices (2021); CZSO (2021); MoH CR (2021a, 2021b).

### 2.2. Methods

As was mentioned in Section 1, the main aim of our research is the identification of the benefits of digitalisation of medical devices in hospitals in COVID-19 times, focusing on a case study of the Czech Republic. This case study presents the equipment/facilities use changes between the years 2019 and 2020 in a selected hospital in Prague and the influence of COVID-19 on the use of these resources. The possible differences in both distribution of resources and economic impact will be discussed.

The following research questions were proposed within the topic:

(RQ1). Is the impact of the COVID-19 pandemic on the distribution of resources and facility management in hospitals significant?

(RQ2). Is the digitalisation of medical devices beneficial for the selected hospital in Prague?

For our research, we used the following methods: literature review, data analysis, cost-benefit analysis, correlation analysis, and regression analysis.

Our case study will present results for the selected hospital in Prague, Czech Republic, which introduced a significant digitalisation tool (new software EFAS, described above) in 2018. The EFAS information system provides facility management for the hospital in

digital form, especially the digitalisation of medical devices. The year 2018 represented the pilot year of the software implementation; therefore, the data are not statistically relevant. For our research, we used data from the years 2019 and 2020.

Correlation analysis (Pearson's correlation coefficient) and regression analysis were carried out based on both the data collected in the selected hospital and official data of the Czech Statistical Office (CZSO 2021) and the Ministry of Health of the Czech Republic (MoH CR 2021a, 2021b) for the COVID-19 period (2020) and the previous year (2019). The possible links and connections between the variables were evaluated. The potential impact of the selected indicators in the Czech Republic was examined. The authors used linear regression models.

The key one is the general regression model MOD which counts the relation between the equipment/facilities use frequency and the number of COVID-19 positive people in the Czech Republic/number of COVID-19 positive people in Prague. The general regression equation is as follows:

$$Y = \beta_0 + \beta_1 X_1 + \mu \tag{1}$$

In this equation, parameters $\beta_0$ and $\beta_1$ represent regression coefficients that reflect the impact of the independent variable on the dependent variable. The dependent variable Y represents equipment/facilities use frequency in the hospital in Prague. The overview of such equipment/facilities used in the analysis is presented in Table 3. The parameter $\mu$ represents a random element of the model. The independent variable $X_1$ in the regression equation is the number of COVID-19 positive people in the Czech Republic/number of COVID-19 positive people in Prague.

**Table 3.** Expected effect on equipment use in hospitals.

| Variable | Role | Expected Effect |
|----------|------|-----------------|
| EQUIP | Dependent | - |
| INF | Dependent | - |
| INFA | Dependent | - |
| LIDI | Dependent | - |
| INJ | Dependent | - |
| DEFL | Dependent | - |
| DEF | Dependent | - |
| BED | Dependent | - |
| BEDE | Dependent | - |
| LUVE | Dependent | - |
| COVP | Key explanatory | Negative |
| COV | Control explanatory | Not clear |

Source: Own processing.

Table 3 shows the overview of all variables and their expected/theoretical effect on the equipment/facilities use in a selected hospital in Prague.

## 3. Results

### 3.1. Temporal Changes in Equipment/Facilities Use

The first area of focus was on the development of COVID-19 positive citizens in the Czech Republic in 2020 (Figure 3).

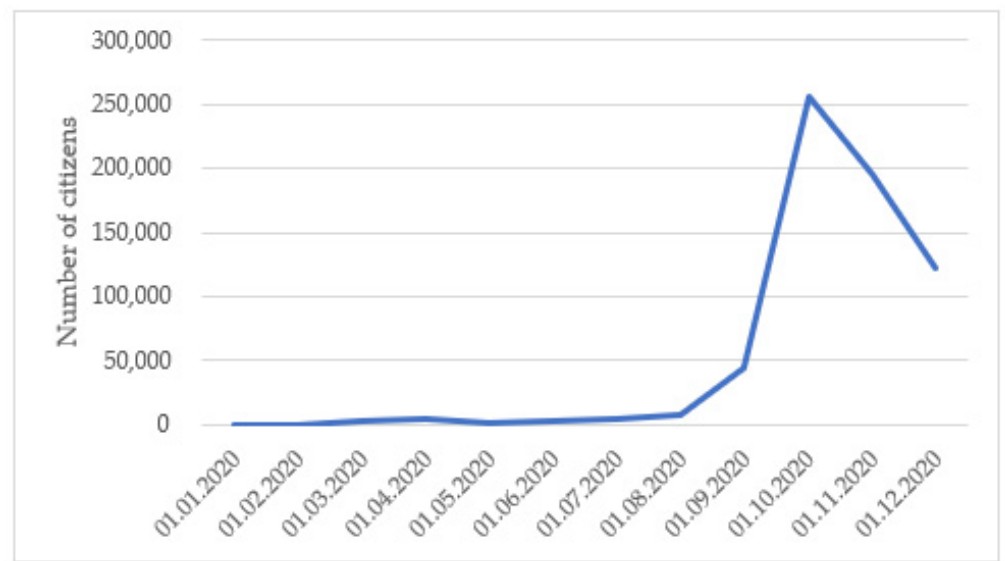

**Figure 3.** COVID-19 positive citizens in the Czech Republic in 2020, monthly data. Source: Own processing, based on the data CZSO (2021); MoH CR (2021a).

Regarding the situation in Prague in 2020, the development of COVID-19 positive citizens was slightly different from the whole Czech Republic, as is visible in Figures 4–6.

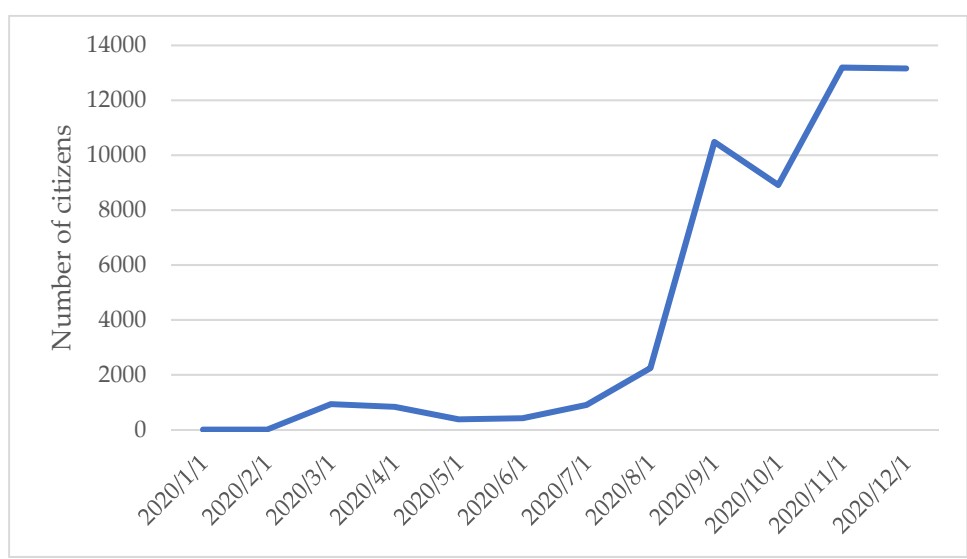

**Figure 4.** COVID-19 positive citizens in Prague in 2020, monthly data. Source: Own processing, based on the data CZSO (2021); MoH CR (2021b).

The increase in the number of COVID-19 positive people started earlier in Prague in 2020 than in the whole Czech Republic. Moreover, we can say that the development of time series and their peaks differ from the whole Czech Republic.

Therefore, for the case study of the selected hospital in Prague, the time series of COVID-19 positive citizens in Prague should be more significant and valuable. Concerning daily data connected with the situation in Prague in 2020, Figure 5 shows the period March–June 2020, and Figure 6 shows the development over the period July–December 2020.

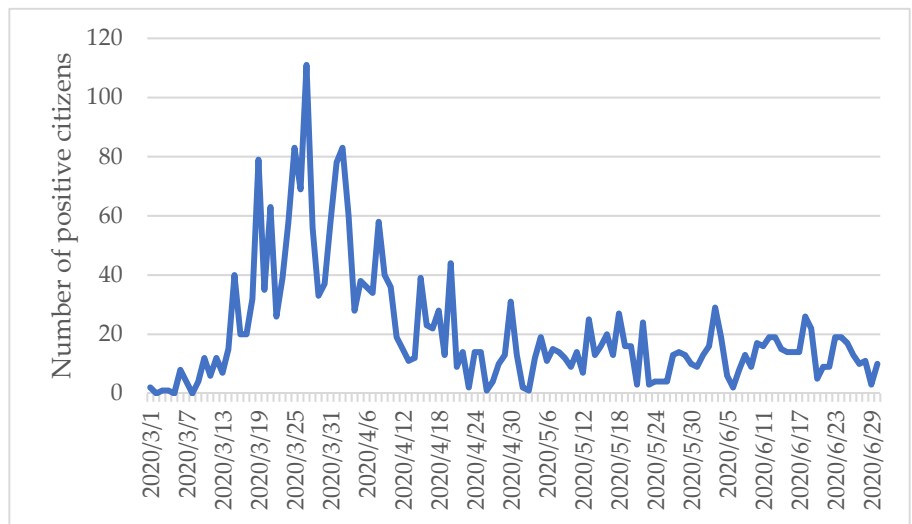

**Figure 5.** Increase in COVID-19 positive citizens in Prague in March–June 2020, daily data. Source: Own processing, based on the data CZSO (2021); MoH CR (2021b).

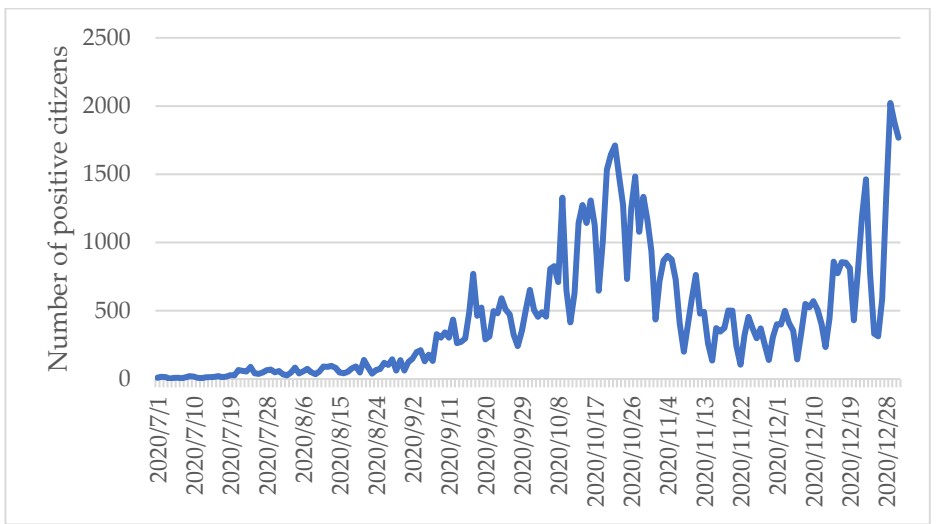

**Figure 6.** Increase in COVID-19 positive citizens in Prague in July–December 2020, daily data. Source: Own processing, based on the data CZSO (2021); MoH CR (2021b).

The next focus was on the aspects of the digitalisation of medical devices in the selected hospital in Prague. As was mentioned before, the new software EFAS was implemented in 2018 (the pilot year). The statistically valuable data are available for years 2019 and 2020. Figure 7 shows the development of selected equipment/facilities use in the years 2019–2020.

For COVID-19 period evaluation, Figure 6 shows equipment/facilities use in 2020 in more detail.

Observing Figures 4–6 and 8, an increase in COVID-19 positive people seems to relate to a decrease in the equipment/facilities use in the hospital. It shows a change in the distribution of resources, a decrease in the use of some equipment/facilities, and the equipment/facilities move between the departments, hospitals, and/or healthcare organizations. It can also show a change in equipment/facilities use and selected equipment move to COVID-19 departments. We will use more sophisticated tools for analysing the equipment/facilities use in the hospital in Prague, such as correlation, and regression analysis, gradually.

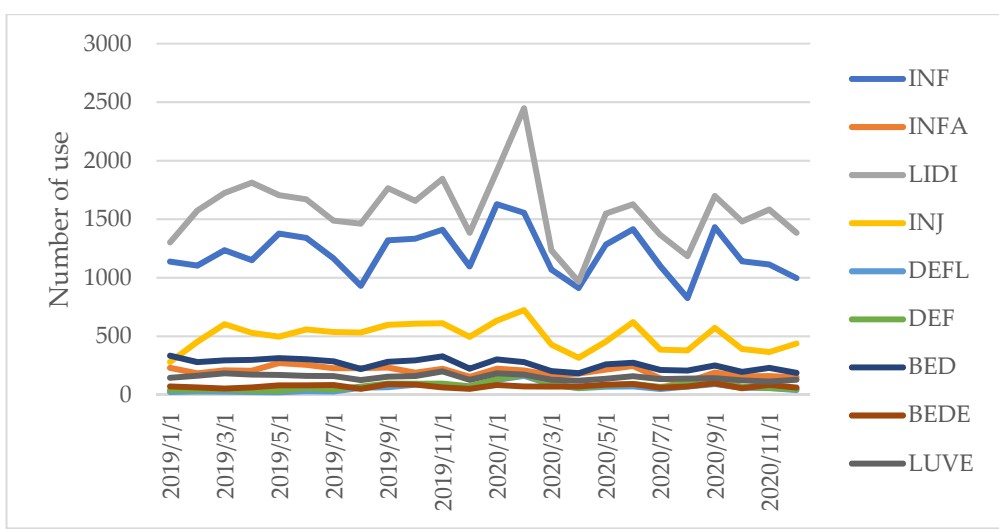

**Figure 7.** Equipment/facilities use in hospital in Prague in 2019–2020. Source: Own processing.

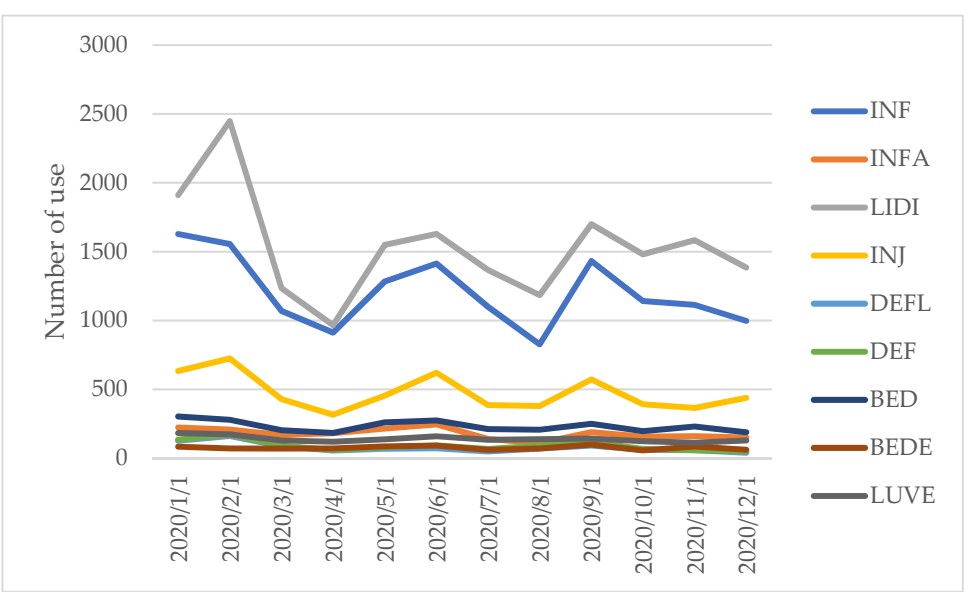

**Figure 8.** Equipment/facilities use in hospital in Prague in 2020. Source: Own processing.

### 3.2. Correlation and Regression Analysis

For the correlation analysis, variables connected with selected equipment/facilities use in the hospital in Prague and COVID-19 positive people in Prague were used. Table 4 shows the results of the correlation analysis.

With exception of elegance bed (BEDE), we can observe a negative correlation between COVID-19 positive people in Prague and equipment/facilities use in the hospital in Prague. Regarding statistical significance, there is a statistically significant correlation coefficient between COVID-19 positive people in Prague (COVP) and infusion pump ARGUS (INFA, −0.46), bed (BED, −0.49), and lung ventilator (LUVE, −0.51).

In the next step, we focused on regression analysis using variables INFA, BED, and LUVE as dependent variables and COVID-19 positive people in Prague as an independent variable. To be sure not to omit any important results, we created more regression models for all variables presented in Table 3, working with both single and multiply regression models. However, only the variables INFA, BED, and LUVE showed statistically significant results in single regression models. Such results are presented in Tables 5 and 6.

**Table 4.** Results of correlation analysis.

| Variables | INF | INFA | LIDI | INJ | DEFL | DEF | BED | BEDE | LUVE | EQUIP | COVP |
|---|---|---|---|---|---|---|---|---|---|---|---|
| INF | 1 | | | | | | | | | | |
| INFA | 0.59 | 1 | | | | | | | | | |
| LIDI | 0.82 | 0.45 | 1 | | | | | | | | |
| INJ | 0.75 | 0.49 | 0.82 | 1 | | | | | | | |
| DEFL | 0.49 | −0.10 | 0.48 | 0.49 | 1 | | | | | | |
| DEF | 0.55 | −0.05 | 0.52 | 0.57 | 0.97 | 1 | | | | | |
| BED | 0.66 | 0.75 | 0.59 | 0.49 | −0.07 | 0.002 | 1 | | | | |
| BEDE | 0.52 | 0.37 | 0.21 | 0.23 | 0.23 | 0.31 | 0.31 | 1 | | | |
| LUVE | 0.68 | 0.57 | 0.69 | 0.70 | 0.12 | 0.19 | 0.83 | 0.11 | 1 | | |
| EQUIP | 0.93 | 0.58 | 0.95 | 0.88 | 0.53 | 0.59 | 0.68 | 0.37 | 0.75 | 1 | |
| COVP | −0.18 | −0.46 | −0.13 | −0.30 | −0.003 | −0.05 | −0.49 | 0.10 | −0.51 | −0.24 | 1 |

Statistical significance $p < 0.05$. Source: Own processing.

**Table 5.** Results of regression analysis.

| | Sig. | Coef. | Sig. | Coef. | Sig. | Coef. | Sig. | Coef. |
|---|---|---|---|---|---|---|---|---|
| Y1—INFA | 0.024 ** | −0.004 | x | x | x | x | x | x |
| Y2—BED | x | x | 0.014 ** | −0.005 | x | x | x | x |
| Y3—LUVE | x | x | x | x | 0.011 ** | −0.003 | x | x |
| Y4—ALL | x | x | x | x | x | x | 0.007 *** | −0.012 |
| VIF | 0.000 | | 0.000 | | 0.000 | | 0.000 | |
| Constant | 207.460 | | 271.524 | | 156.122 | | 635.106 | |
| Observ. | 310 | | 310 | | 310 | | 310 | |
| R2 | 0.460 | | 0.493 | | 0.509 | | 0.533 | |
| Signif. | 0.024 ** | | 0.014 ** | | 0.011 ** | | 0.007 *** | |
| DW | 1.769 | | 1.642 | | 1.920 | | 1.883 | |

Statistical significance *** $p < 0.01$, ** $p < 0.05$, * $p < 0.1$. Source: Own processing.

**Table 6.** Regression equations.

| Model | Regression Equation |
|---|---|
| MOD1 | $Y = 207.460 − 0.004\, x_1 + u$ |
| MOD2 | $Y = 271.524 − 0.005\, x_1 + u$ |
| MOD3 | $Y = 156.122 − 0.003\, x_1 + u$ |
| MOD4 | $Y = 635.106 − 0.012\, x_1 + u$ |

Source: Own processing.

Based on all regression models and equations, we can say that an increase in the number of COVID-19 positive people influenced a change/decrease in equipment/facilities use, such as infusion pump ARGUS, bed, and lung ventilator.

### 3.3. Cost-Benefit Analysis of Digitalisation of Medical Devices in Hospitals

It is not easy to identify, moreover to calculate, possible benefits and costs of digitalisation of medical devices in hospitals. In the previous chapter, the results of correlation and regression analysis show us that COVID-19 influenced the distribution of processes and equipment/facilities use in the hospital in Prague. It is connected both with the economic and financial aspects of management. However, COVID-19 is only one example of possible changes in hospital processes and facility management. It is necessary for all managers to have updated information for their decision-making.

It should be underlined that cost-benefit analysis of digitalisation of medical devices in hospitals is connected with various periods. We should distinguish between short-time and long-time costs and benefits.

Regarding the phase of implementation of digitalisation and new software in hospitals, it is clear that there are costs connected with system infrastructure, systems applications,

office administration, scanning of paper documents, technical support, and training of management and employees. The overview of such costs is described in Table 7.

**Table 7.** Cost-Benefit Analysis—Costs identification.

| Areas | Possible Costs Identification |
|---|---|
| System infrastructure | System infrastructure costs—SW and HW; acquisition and maintenance costs |
| System applications | Development and maintenance costs |
| Office administration | PCs, monitors, printers, scanners; acquisition and maintenance costs |
| Scanning of paper documents | The cost of scanning existing documentation and converting materials online |
| Technical support | Costs connected with professionals—technicians |
| Management and employees | Costs of initial and ongoing training of management and responsible employees |

Source: Own processing.

Most of these costs occur in a short time period. On the other hand, a long time period is connected more with possible benefits. These benefits are represented by various kinds of costs reductions and also with additional income. The overview of possible cost reduction is described in Table 8.

**Table 8.** Cost-Benefit Analysis—Benefits identification—Cost reduction.

| Areas | Possible Benefits Identification—Cost Reduction |
|---|---|
| Time management | Time optimisation in equipment/facilities use |
| Paper documentation | Reduction in costs connected with storage boxes, cabinets, folders |
| The storage space | Reduced storage space |
| Management | Optimisation of processes |
| Employees | Reduced personnel and staff costs, optimisation of number of employees |
| Materials | Reduction in consumption of paper documents. Reduction of delivery materials |
| Orders to the departments | Minimize duplicate orders, order sets (quantity discount) |
| Services, outsourcing | Optimization of internal/external provision of services, cost reduction |

Source: Own processing.

Generally, reduction in costs is connected with various processes and materials in the hospital, such as paper documentation, storage space, management, time management, employees, materials, orders to the departments, and services.

Besides cost reductions, digitalisation also helps to improve the income side of hospitals. The overview of possible additional income is described in Table 9.

**Table 9.** Cost-Benefit Analysis—Benefits identification—Additional income.

| Areas | Possible Benefits Identification—Additional Income |
|---|---|
| Modified storage space | Additional annual income from the rooms |
| Modified occasional storage space | Additional annual income from the rooms |
| Medical records | Additional income from more served patients—speeding up medical records |

Source: Own processing.

Free storage space (both regular and occasional) can be modified and adapted to other rooms. Moreover, hospitals can rent these rooms. Based on our case study, the hospital rented additional free rooms to providers of services connected with healthcare, such as massage salons, beauty salons, and/or nutritional counselling.

Focusing on the impact of digitalisation of medical devices in hospitals in time, based on our research, we can say that the costs occur mostly in a short time period and the benefits in a long time period.

## 4. Discussion

Digitalisation is one of the key tools of modern management. All its aspects now face the challenges of the post-COVID-19 period, and the process of digitalisation is very important in this development. Customization, flexibility, acceleration of all processes, and pressure to constantly streamline HR management are some of the features of modern management (Sternad Zabukovšek et al. 2021). Digitalisation permeates all spheres of business and should be an integral part of the component (active tool) of healthcare management. Generally, digital transformation relates to various fields of economic entities, including hospitals and health care facilities.

Regarding the first research question (RQ1)—"Is the impact of the COVID-19 pandemic on the distribution of resources and facility management in hospitals significant?", based on our research the answer is "rather yes". Concerning the literature review and case study data, COVID-19 caused a great acceleration in the use of technology, digitalisation of processes, and new forms of working. Moreover, COVID-19 represents a huge impetus for innovation in many sectors, including healthcare. As Fabiano (2020) highlighted: "Like a rocket blasting off for the moon, the global pandemic accelerated 20 years of pent-up innovation for the healthcare industry into 8 months." We agree with Ageron et al. (2020) and Tortorella et al. (2021) that this period is characterised by a complex and dynamic environment in hospitals, where digitalisation affects several aspects of life, such as internal logistics distribution and facility management.

A new system of distribution and connected facility management relates to financial issues. On the one side, the healthcare system needs more financial and other resources for covering all necessary medical products and services. On the other side, there is pressure on the effectiveness and optimisation of resources in hospitals and healthcare facilities. All resources are optimized—employees, medical products, medical services, rooms, and capacities. Economic optimisation represents the challenge for digitalisation and the new management approach.

Dealing with the second research question (RQ2)—"Is the digitalisation of medical devices beneficial for the selected hospital in Prague?", based on our research the answer is not so clear. Focusing on the impact of digitalisation of medical devices in hospitals in time, based on our research, we can say that the costs occur mostly in a short time period and the benefits over a long-time period. However, it is not easy to identify all possible savings, as some of them only have a non-financial expression. Therefore, the answer is "rather yes", but only over a long-time period.

Our results correspond with the study performed by Choi et al. (2013). The authors analysed the economic effects of an electronic medical record system in hospitals that used cost-benefit analysis based on the differential costs of managerial accounting. The benefits included both cost reductions and additional revenues.

As Dasgupta and Narendran (2021) underline, modern hospitals should be technologically sophisticated healthcare facilities with technologically specialised personnel. On the other hand, healthcare is a multidisciplinary sector and various professionals from different fields should cooperate. New solutions and ideas can emerge in interdisciplinary teams, using disruptive and out-of-box thinking.

Regarding this point, we agree with Ageron et al. (2020) that researchers should focus in more detail on the issues related to the digitalisation of public supply chains, such as public hospitals supply chains, which are under-studied. Innovations and developments in health technology can contribute significantly to the quality of health care provided by various health facilities, but have also brought new challenges in the management of health care services. Health service planners, hospital administrators, physicians, and other health

care professionals need to understand the forces that add value to the cost-effectiveness and efficiency of health care delivery systems.

Concerning further research, firstly, 2021 should be examined. This "second COVID-19 year" has specific characteristics, such as vaccination, new tools in hospitals, and a new managerial approach. Simultaneously, the economic impacts of COVID-19 are increasing. Secondly, qualitative research can help to add the personal view of employees and managers. The presented analysis is based on a dataset extracted from system EFAS. Expert interviews with managers can represent valuable input to the overall picture of the hospital.

## 5. Conclusions

Generally, COVID-19 has been a huge accelerator for R&D, innovation, optimization, and new ideas. Technology and health care equipment/facilities play a significant role in health care services. Therefore, effective facility management can contribute to the improvement of healthcare.

The main aim of our research was the identification of the benefits of digitalisation of medical devices in hospitals in COVID-19 times, focusing on a case study in the Czech Republic. Our methodological approach included a literature review, data analysis, correlation analysis, and regression analysis. The case study presented the changes of the equipment/facilities use in 2019 and 2020 in a selected hospital in Prague and the impact of COVID-19 on the use of these resources. Management and financial issues were discussed. Based on the results, COVID-19 influenced both the distribution of resources and facility management in hospitals. Economic benefits are represented mainly by various kinds of savings and optimization of both processes and employees.

It is clear that digitalisation represents an important source of information for management in hospitals. Based on the presented research, we can recommend digitalisation as a suitable tool for managers and headquarters in hospitals. They can easily observe all processes and persons and optimize the healthcare services provided.

**Author Contributions:** Conceptualization, J.Z.; methodology, J.Z.; software, L.P.; validation, E.C.; formal analysis, E.C.; investigation, L.P.; resources, L.P.; data curation, E.C.; writing—original draft preparation, J.Z.; writing—review and editing, L.P.; visualization, J.Z.; supervision, J.Z.; project administration, E.C.; funding acquisition, J.Z. All authors have read and agreed to the published version of the manuscript.

**Funding:** This research was funded by European Commission, project EFAS Season One CZ.01.1.02/0.0/0.0/19_274/0018/365, project ROKA, number CZ.02.2.69/0.0/0.0/18_054/0014592, and project EFAS Season Two CZ.01.1.02/0.0/0.0/20_321/0024495.

**Institutional Review Board Statement:** Not applicable.

**Informed Consent Statement:** Not applicable.

**Data Availability Statement:** Data supporting reported results are available after request.

**Conflicts of Interest:** The authors declare no conflict of interest.

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
