# Peer review of "Digitalisation in Hospitals in COVID-19 Times—A Case Study of the Czech Republic"

_economies, doi:10.3390/economies10030068_

Round 1

Reviewer 1 Report

Dear authors, 

the work you presented is well documented and executed and I just suggest two minor corrections only regarding the readability: 

  1. On page 5, you introduce the research questions and you're referring to this questions on page 13. On page 13, the acronyms RQ1 and RQ2 are introduced. I would suggest you introduce the acronyms already on page 5.
  2. In my opinion, Figure 9 is rather useless because it lacks readability in the current form. Graph visualizations are fancy, but only when you can interact with the visualization, which is not possible here. Therefore, I suggest to completely remove figure 9, since it's overall relevance to your article is rather low.  

Author Response

Dear reviewer, we are very thankful to you for your time invested in reading/reviewing our paper and the suggestions and comments you have given to our previous version of the paper. Addressing these suggestions in the revised version has definitely improved the paper. 

Dear reviewers,

Thank you for your comments and (positive) feedback, as well as for suggestions on how to improve our paper. They were truly helpful and motivating, having made the revised version of our paper even better than the original one.

The work you presented is well documented and executed and I just suggest two minor corrections only regarding the readability: 

Thank you for your positive feedback.

  1. On page 5, you introduce the research questions and you're referring to this questions on page 13. On page 13, the acronyms RQ1 and RQ2 are introduced. I would suggest you introduce the acronyms already on page 5.

Thank you for your comment. We introduced the acronyms already on page 5.

  1. In my opinion, Figure 9 is rather useless because it lacks readability in the current form. Graph visualizations are fancy, but only when you can interact with the visualization, which is not possible here. Therefore, I suggest to completely remove figure 9, since it's overall relevance to your article is rather low.  

Thank you for your comment. We removed Figure 9.

Reviewer 2 Report

The topic of the article is very interesting and very relevant due to the current reference (management of the Covid 19 crisis through the application of digital solutions in the healthcare sector). 

In principle, it would be good not to speak of digitalisation as a whole, but of digital solutions that enable concrete process optimisations in the healthcare sector. Digitalisation is a buzz word, which should be specified. There should be a little more emphasis on narrowing down digitalisation. Is it about AI-based diagnostics or is it about medical devices in the inpatient environment? 

Overall, and this is probably the main criticism, it is not entirely clear whether we are dealing with spurious correlations or actual correlations of digitalisation and its influence on effectiveness and optimisation of sources in hospitals and healthcare facilities. The empirical data and calculations are certainly interesting, but due to the vague definition and the case study character, correlation and causality do not necessarily lie side by side. The authors may have answers to my question and can certainly use my comments to offer more clarity here. 

Otherwise, case studies like this one from the Czech Republic are important contributions to a substantive debate on digitalisation effects in the clinical setting. Therefore, I would welcome the publication if work is done on the paper's strength and clarity beforehand. 

Author Response

Dear reviewer, we are very thankful to you for your time invested in reading/reviewing our paper and the suggestions and comments you have given to our previous version of the paper. Addressing these suggestions in the revised version has definitely improved the paper. 

The topic of the article is very interesting and very relevant due to the current reference (management of the Covid 19 crisis through the application of digital solutions in the healthcare sector). 

Thank you for your positive feedback.

In principle, it would be good not to speak of digitalisation as a whole, but of digital solutions that enable concrete process optimisations in the healthcare sector. Digitalisation is a buzz word, which should be specified. There should be a little more emphasis on narrowing down digitalisation. Is it about AI-based diagnostics or is it about medical devices in the inpatient environment? 

Thank you for your comment. We modified the main aim of our research as follows:

The main aim of our research was the identification of the benefits of digitalisation of medical devices in hospitals in COVID-19 times, focusing on a case study of the Czech Republic.

It was modified in the following chapters: Abstract, Methods (2.2.), Conclusions.

Overall, and this is probably the main criticism, it is not entirely clear whether we are dealing with spurious correlations or actual correlations of digitalisation and its influence on effectiveness and optimisation of sources in hospitals and healthcare facilities. The empirical data and calculations are certainly interesting, but due to the vague definition and the case study character, correlation and causality do not necessarily lie side by side. The authors may have answers to my question and can certainly use my comments to offer more clarity here. 

Thank you for your comment. In the case study, we find an actual correlation between the number of COVID positive people and the change in the use of medical devices. We can see that digitisation of medical devices allows us to monitor this change, and it is possible to optimise the use of medical devices.

There is no direct relationship/causality between efficiency and digitisation of medical devices. On the other hand, based on the results, we can see that digitalisation of medical devices is an important tool for hospital management.

The case study presents the equipment/facilities use changes between the years 2019 and 2020 in a selected hospital in Prague and the influence of COVID-19 on such use of sources. The possible differences in both distributions of sources and economic impact are discussed.

Based on your comments and suggestions, we modified the text of the paper, to be more clear and focused.

There is the overview of modifications and changes:

Abstract

  • Row 16,17: digitalisation of medical devices

2.2. Methods

  • Row 187: digitalisation of medical devices
  • Row 195 – modification of Research question: „RQ2) Is the digitalisation of medical devices beneficial for a selected hospital in Prague?“
  • Row 201-202 – we added a new sentence connected with the case study: „The EFAS information system provides facility management for the hospital in digital form, especially the digitalisation of medical devices.“
  1. Results
  • Row 266: digitalization of medical devices
  • Row 318, 319, 320: digitalisation of medical devices
  • Headline 3.3. – we added digitalisation of medical devices
  • Row 326: digitalisation of medical devices
  • Row 355: digitalisation of medical devices
  1. Discussion
  • Row 385, 386: digitalisation of medical devices
  • Row 387: digitalisation of medical devices
  • We removed Figure 9.

Conclusions

  • Row 421, 422: digitalisation of medical devices

Otherwise, case studies like this one from the Czech Republic are important contributions to a substantive debate on digitalisation effects in the clinical setting. Therefore, I would welcome the publication if work is done on the paper's strength and clarity beforehand. 

Thank you for your positive feedback.

Dear reviewers, we are very thankful to you for your time invested in reading/reviewing our paper and the suggestions and comments you have given to our previous version of the paper. Addressing these suggestions in the revised version has definitely improved the paper.

Round 2

Reviewer 2 Report

Thank you. All recommendations were taken into account in an adequate form.